# Designing smoothing functions for improved worst-case competitive ratio in online optimization

**Reza Eghbali**
Department of Electrical Engineering
University of Washington
Seattle, WA 98195
eghbali@uw.edu

**Maryam Fazel**
Department of Electrical Engineering
University of Washington
Seattle, WA 98195
mfazel@uw.edu

## Abstract

Online optimization covers problems such as online resource allocation, online bipartite matching, adwords (a central problem in e-commerce and advertising), and adwords with separable concave returns. We analyze the worst case competitive ratio of two primal-dual algorithms for a class of online convex (conic) optimization problems that contains the previous examples as special cases defined on the positive orthant. We derive a sufficient condition on the objective function that guarantees a constant worst case competitive ratio (greater than or equal to $\frac{1}{2}$) for monotone objective functions. We provide new examples of online problems on the positive orthant that satisfy the sufficient condition. We show how smoothing can improve the competitive ratio of these algorithms, and in particular for separable functions, we show that the optimal smoothing can be derived by solving a convex optimization problem. This result allows us to directly optimize the competitive ratio bound over a class of smoothing functions, and hence *design* effective smoothing customized for a given cost function.

## 1 Introduction

Given a proper convex cone $K \subset \mathbb{R}^n$, let $\psi : K \mapsto \mathbb{R}$ be an upper semi-continuous concave function. Consider the optimization problem

$$
\begin{array}{ll}
\text{maximize} & \psi\left(\sum_{t=1}^m A_t x_t\right) \\
\text{subject to} & x_t \in F_t, \qquad \forall t \in [m],
\end{array}
\tag{1}
$$

where for all $t \in [m] := \{1, 2, \ldots, m\}$, $x_t \in \mathbb{R}^l$ are the optimization variables and $F_t$ are compact convex constraint sets. We assume $A_t \in \mathbb{R}^{n \times l}$ maps $F_t$ to $K$; for example, when $K = \mathbb{R}^n_+$ and $F_t \subset \mathbb{R}^l_+$, this assumption is satisfied if $A_t$ has nonnegative entries. We consider problem (1) in the online setting, where it can be viewed as a sequential game between a player (online algorithm) and an adversary. At each step $t$, the adversary reveals $A_t$, $F_t$ and the algorithm chooses $\hat{x}_t \in F_t$. The performance of the algorithm is measured by its competitive ratio, i.e., the ratio of objective value at $\hat{x}_1, \ldots, \hat{x}_m$ to the offline optimum. Problem (1) covers (convex relaxations of) various online combinatorial problems including online bipartite matching [14], the "adwords" problem [16], and the secretary problem [15]. More generally, it covers online linear programming (LP) [6], online packing/covering with convex cost [3, 4, 7], and generalization of adwords [8]. In this paper, we study the case where $\partial \psi(u) \subset K^*$ for all $u \in K$, i.e., $\psi$ is monotone with respect to the cone $K$.

The competitive performance of online algorithms has been studied mainly under the worst-case model (e.g., in [16]) or stochastic models (e.g., in [15]). In the worst-case model one is interested in lower bounds on the competitive ratio that hold for any $(A_1, F_1), \ldots, (A_m, F_m)$. In stochastic models, adversary choses a probability distribution from a family of distributions to generate

$(A_1, F_1), \ldots, (A_m, F_m)$, and the competitive ratio is calculated using the expected value of the algorithm's objective value. Online bipartite matching and its generalization, the "adwords" problem, are the two main problems that have been studied under the worst case model. The greedy algorithm achieves a competitive ratio of $1/2$ while the optimal algorithm achieves a competitive ratio of $1 - 1/e$ (as bid to budget ratio goes to zero) [16, 5, 14, 13]. A more general version of Adwords in which each agent (advertiser) has a concave cost has been studied in [8].

The majority of algorithms proposed for the problems mentioned above rely on a primal-dual framework [5, 6, 3, 8, 4]. The differentiating point among the algorithms is the method of updating the dual variable at each step, since once the dual variable is updated the primal variable can be assigned using a simple complementarity condition. A simple and efficient method of updating the dual variable is through a first order online learning step. For example, the algorithm stated in [9] for online linear programming uses mirror descent with entropy regularization (multiplicative weight updates algorithm) once written in the primal dual language. Recently, the work in [9] was independently extended to random permutation model in [12, 2, 11]. In [2], the authors provide competitive difference bound for online convex optimization under random permutation model as a function of the regret bound for the online learning algorithm applied to the dual.

In this paper, we consider two versions of the greedy algorithm for problem (1), a sequential update and a simultaneous update algorithm. The simultaneous update algorithm, Algorithm 2, provides a direct saddle-point representation of what has been described informally in the literature as "continuous updates" of primal and dual variables. This saddle point representation allows us to generalize this type of updates to non-smooth function. In section 2, we bound the competitive ratios of the two algorithms. A sufficient condition on the objective function that guarantees a non-trivial worst case competitive ratio is introduced. We show that the competitive ratio is at least $\frac{1}{2}$ for a monotone non-decreasing objective function. Examples that satisfy the sufficient condition (on the positive orthant and the positive semidefinite cone) are given. In section 3, we derive optimal algorithms, as variants of greedy algorithm applied to a smoothed version of $\psi$. For example, Nesterov smoothing provides optimal algorithm for the adwords problem. The main contribution of this paper is to show how one can derive the optimal smoothing function (or from the dual point of view the optimal regularization function) for separable $\psi$ on positive orthant by solving a convex optimization problem. This gives an implementable algorithm that achieves the optimal competitive ratio derived in [8]. We also show how this convex optimization can be modified for the design of smoothing function specifically for the sequential algorithm. In contrast, [8] only considers continuous updates.

The algorithms considered in this paper and their general analysis are the same as those we considered in [10]. In [10], the focus is on non-monotone functions and online problems on the positive semidefinite cone. However, the focus of this paper is on monotone functions on the positive orthant. In [10], we only considered Nesterov smoothing and only derived competitive ratio bounds for the simultaneous algorithm.

**Notation.** Given a function $\psi : \mathbb{R}^n \mapsto \mathbb{R}$, $\psi^*$ denotes the concave conjugate of $\psi$ defined as $\psi^*(y) = \inf_u \langle y, u \rangle - \psi(u)$, for all $y \in \mathbb{R}^n$. For a concave function $\psi$, $\partial \psi(u)$ denotes the set of supergradients of $\psi$ at $u$, i.e., the set of all $y \in \mathbb{R}^n$ such that $\forall u' \in \mathbb{R}^n : \quad \psi(u') \le \langle y, u' - u \rangle + \psi(u)$. The set $\partial \psi$ is related to the concave conjugate function $\psi^*$ as follows. For an upper semi-continuous concave function $\psi$ we have $\partial \psi(u) = \operatorname{argmin}_y \langle y, u \rangle - \psi^*(y)$. A differentiable function $\psi$ has a Lipschitz continuous gradient with respect to $\|\cdot\|$ with continuity parameter $1/\mu > 0$ if for all $u, u' \in \mathbb{R}^n$, $\|\nabla \psi(u') - \nabla \psi(u)\|^* \le 1/\mu \|u - u'\|$, where $\|\cdot\|^*$ is the dual norm to $\|\cdot\|$.

The dual cone $K^*$ of a cone $K \subset \mathbb{R}^n$ is defined as $K^* = \{y \mid \langle y, u \rangle \ge 0 \ \forall u \in K\}$. Two examples of self-dual cones are the positive orthant $\mathbb{R}^n_+$ and the cone of $n \times n$ positive semidefinite matrices $S^n_+$. A proper cone (pointed convex cone with nonempty interior) $K$ induces a partial ordering on $\mathbb{R}^n$ which is denoted by $\le_K$ and is defined as $x \le_K y \Leftrightarrow y - x \in K$.

## 1.1 Two primal-dual algorithms

The (Fenchel) dual problem for problem (1) is given by

$$\text{minimize} \quad \sum_{t=1}^m \sigma_t(A_t^T y) - \psi^*(y), \tag{2}$$

where the optimization variable is $y \in \mathbb{R}^n$, and $\sigma_t$ denotes the *support function* for the set $F_t$ defined as $\sigma_t(z) = \sup_{x \in F_t} \langle x, z \rangle$. A pair $(x^*, y^*) \in (F_1 \times \ldots \times F_m) \times K^*$ is an optimal primal-dual pair if and only if

$$x_t^* \in \operatorname*{argmax}_{x \in F_t} \langle x, A_t^T y^* \rangle, \quad y^* \in \partial \psi(\sum_{t=1}^m A_t x_t^*), \qquad \forall t \in [m].$$

Based on these optimality conditions, we consider two algorithms. Algorithm 1 updates the primal and dual variables *sequentially*, by maintaining a dual variable $\hat{y}_t$ and using it to assign $\hat{x}_t \in \operatorname{argmax}_{x \in F_t} \langle x, A_t^T \hat{y}_t \rangle$. The then algorithm updates the dual variable based on the second optimality condition. By the assignment rule, we have $A_t \hat{x}_t \in \partial \sigma_t(\hat{y}_t)$, and the dual variable update can be viewed as $\hat{y}_{t+1} \in \operatorname{argmin}_y \langle \sum_{s=1}^t A_s \hat{x}_s, y \rangle - \psi^*(y)$. Therefore, the dual update is the same as the update in dual averaging [18] or Follow The Regularized Leader (FTRL) [20, 19, 1] algorithm with regularization $-\psi^*(y)$.

---

**Algorithm 1** Sequential Update

---

Initialize $\hat{y}_1 \in \partial \psi(0)$
**for** $t \leftarrow 1$ to $m$ **do**
    Receive $A_t, F_t$
    $\hat{x}_t \in \operatorname{argmax}_{x \in F_t} \langle x, A_t^T \hat{y}_t \rangle$
    $\hat{y}_{t+1} \in \partial \psi(\sum_{s=1}^t A_s \hat{x}_s)$
**end for**

---

Algorithm 2 updates the primal and dual variables *simultaneously*, ensuring that

$$\tilde{x}_t \in \operatorname*{argmax}_{x \in F_t} \langle x, A_t^T \tilde{y}_t \rangle, \qquad \tilde{y}_t \in \partial \psi(\sum_{s=1}^t A_s \tilde{x}_s).$$

This algorithm is inherently more complicated than algorithm 1, since finding $\tilde{x}_t$ involves solving a saddle-point problem. This can be solved by a first order method like mirror descent algorithm for saddle point problems. In contrast, the primal and dual updates in algorithm 1 solve two separate maximization and minimization problems [1].

---

**Algorithm 2** Simultaneous Update

---

**for** $t \leftarrow 1$ to $m$ **do**
    Receive $A_t, F_t$
    $(\tilde{y}_t, \tilde{x}_t) \in \arg\min_y \max_{x \in F_t} \langle y, A_t x + \sum_{s=1}^{t-1} A_s \tilde{x}_s \rangle - \psi^*(y)$
**end for**

---

## 2    Competitive ratio bounds and examples for $\psi$

In this section, we derive bounds on the competitive ratios of Algorithms 1 and 2 by bounding their respective duality gaps. We begin by stating a sufficient condition on $\psi$ that leads to non-trivial competitive ratios, and we assume this condition holds in the rest of the paper. Roughly, one can interpret this assumption as having "diminishing returns" with respect to the ordering induced by a cone. Examples of functions that satisfy this assumption will appear later in this section.

**Assumption 1** *Whenever $u \geq_K v$, there exists $y \in \partial \psi(u)$ that satisfies $y \leq_{K^*} z$ for all $z \in \partial \psi(v)$.*

When $\psi$ is differentiable, assumption 1 simplifies to $u \geq_K v \Rightarrow \nabla \psi(u) \leq_{K^*} \nabla \psi(v)$. That is, the gradient, as a map from $\mathbb{R}^n$ (equipped with $\leq_K$) to $\mathbb{R}^n$ (equipped with $\leq_{K^*}$), is order-reversing. When $\psi$ is twice differentiable, assumption 1 is equivalent to $\langle w, \nabla^2 \psi(u)v \rangle \leq 0$, for all $u, v, w \in K$. For example, this is equivalent to Hessian being element-wise non-positive when $K = \mathbf{R}_+^n$.

Let define $\tilde{y}_{m+1}$ to be the minimum element in $\partial \psi(\sum_{t=1}^m A_t \tilde{x}_t)$ with respect to ordering $\leq_{K^*}$ (such an element exists in the superdifferential by Assumption (1)). Let $P_{\text{seq}} = \psi\left(\sum_{t=1}^m A_t \hat{x}_t\right)$ and $P_{\text{sim}} = \psi\left(\sum_{t=1}^m A_t \tilde{x}_t\right)$ denote the primal objective values for the primal solution produced by the algorithms

1 and 2, and $D_{\text{seq}} = \sum_{t=1}^{m} \sigma_t(A_t^T \hat{y}_t) - \psi^*(\hat{y}_{m+1})$ and $D_{\text{sim}} = \sum_{t=1}^{m} \sigma_t(A_t^T \tilde{y}_t) - \psi^*(\tilde{y}_{m+1})$ denote the corresponding dual objective values.

The next lemma provides a lower bound on the duality gaps of both algorithms.

**Lemma 1** *The duality gaps for the two algorithms can be lower bounded as*

$$P_{\text{sim}} - D_{\text{sim}} \geq \psi^*(\tilde{y}_{m+1}) + \psi(0), \quad P_{\text{seq}} - D_{\text{seq}} \geq \psi^*(\hat{y}_{m+1}) + \psi(0) + \sum_{t=1}^{m} \langle A_t \hat{x}_t, \hat{y}_{t+1} - \hat{y}_t \rangle$$

*Furthermore, if $\psi$ has a Lipschitz continuous gradient with parameter $1/\mu$ with respect to $\|\cdot\|$,*

$$P_{\text{seq}} - D_{\text{seq}} \geq \psi^*(\hat{y}_{m+1}) + \psi(0) - \frac{1}{2\mu} \sum_{t=1}^{m} \|A_t \hat{x}_t\|^2. \tag{3}$$

Note that right hand side of (3) is exactly the regret bound of the FTRL algorithm (with a negative sign) [19]. The proof is given in the appendix. To simplify the notation in the rest of the paper, we assume $\psi(0) = 0$ by replacing $\psi(u)$ with $\psi(u) - \psi(0)$. To quantify the competitive ratio of the algorithms, we define $\alpha_\psi$ as

$$\alpha_\psi = \sup \{ c \mid \psi^*(y) \geq c\psi(u), \ y \in \partial\psi(u), \ u \in K \}, \tag{4}$$

Since $\psi^*(y) + \psi(u) = \langle y, u \rangle$ for all $y \in \partial\psi(u)$, $\alpha_\psi$ is equivalent to

$$\alpha_\psi = \sup\{ c \mid \langle y, u \rangle \geq (c+1)\psi(u), \ y \in \partial\psi(u) \ u \in K \}. \tag{5}$$

Note that $-1 \leq \alpha_\psi \leq 0$, since for any $u \in K$ and $y \in \partial\psi(u)$, by concavity of $\psi$ and the fact that $y \in K^*$, we have $0 \leq \langle y, u \rangle \leq \psi(u) - \psi(0)$. If $\psi$ is a linear function then $\alpha_\psi = 0$, while if $0 \in \partial\psi(u)$ for some $u \in K$, then $\alpha_\psi = -1$.

The next theorem provides lower bounds on the competitive ratio of the two algorithms.

**Theorem 1** *If Assumption 1 holds, we have*

$$P_{\text{sim}} \geq \frac{1}{1 - \alpha_\psi} D^\star, \quad P_{\text{seq}} \geq \frac{1}{1 - \alpha_\psi}(D^\star + \sum_{t=1}^{m} \langle A_t \hat{x}_t, \hat{y}_{t+1} - \hat{y}_t \rangle)$$

*where $D^\star$ is the dual optimal objective. If $\psi$ has a Lipschitz continuous gradient with parameter $1/\mu$ with respect to $\|\cdot\|$,*

$$P_{\text{seq}} \geq \frac{1}{1 - \alpha_\psi}(D^\star - \frac{1}{2\mu} \sum_{t=1}^{m} \|A_t \hat{x}_t\|^2). \tag{6}$$

**Proof:** Consider the simultaneous update algorithm. We have $\sum_{s=1}^{t} A_s \tilde{x}_s \leq_K \sum_{s=1}^{m} A_s \tilde{x}_s$ for all $t$, since $A_s F_s \subset K$ for all $s$. Since $\tilde{y}_t \in \partial\psi(\sum_{s=1}^{t} A_s \tilde{x}_s)$ and $\tilde{y}_{m+1}$ was picked to be the minimum element in $\partial\psi(\sum_{s=1}^{m} A_s \tilde{x}_s)$ with respect to $\leq_{K^*}$, by Assumption 1, we have $\tilde{y}_t \geq_{K^*} \tilde{y}_{m+1}$. Since $A_t x \in K$ for all $x \in F_t$, we get $\langle A_t x, \tilde{y}_t \rangle \geq \langle A_t x, \tilde{y}_{m+1} \rangle$; therefore, $\sigma_t(A_t^T \tilde{y}_t) \geq \sigma_t(A_t^T \tilde{y}_m)$. Thus

$$D_{\text{sim}} = \sum_{t=1}^{m} \sigma_t(A_t^T \tilde{y}_t) - \psi^*(\tilde{y}_m) \geq \sum_{t=1}^{m} \sigma_t(A_t^T \tilde{y}_{m+1}) - \psi^*(\tilde{y}_m) \geq D^\star.$$

Now Lemma 1 and definition of $\alpha_\psi$ give the desired result. The proof for Algorithm 1 follows similar steps. $\qquad\square$

We now consider examples of $\psi$ that satisfy Assumption 1 and derive lower bound on $\alpha_\psi$ for those examples.

**Examples on positive orthant.** Let $K = \mathbb{R}_+^n$ and note that $K^* = K$. To simplify the notation we use $\leq$ instead of $\leq_{\mathbb{R}_+^n}$. Assumption 1 is satisfied for a twice differentiable function if and only if the Hessian is element-wise non-positive over $\mathbb{R}_+^n$. If $\psi$ is separable, i.e., $\psi(u) = \sum_{i=1}^{n} \psi_i(u_i)$, Assumption 1 is satisfied since by concavity for each $\psi_i$ we have $\partial\psi_i(u_i) \leq \partial\psi_i(v_i)$ when $u_i \leq v_i$.

In the basic adwords problem, for all $t$, $F_t = \{x \in \mathbb{R}_+^l \mid \mathbf{1}^T x \leq 1\}$, $A_t$ is a diagonal matrix with non-negative entries, and

$$\psi(u) = \sum_{i=1}^{n} u_i - \sum_{i=1}^{n} (u_i - 1)_+, \tag{7}$$

where $(\cdot)_+ = \max\{\cdot, 0\}$. In this problem, $\psi^*(y) = \mathbf{1}^T(y - \mathbf{1})$. Since $0 \in \partial\psi(\mathbf{1})$ we have $\alpha_\psi = -1$ by (5); therefore, the competitive ratio of algorithm 2 is $\frac{1}{2}$. Let $r = \max_{t,i,j} A_{t,i,j}$, then we have $\sum_{t=1}^m \langle A_t\hat{x}_t, \hat{y}_{t+1} - \hat{y}_t \rangle \le nr$. Therefore, the competitive ratio of algorithm 1 goes to $\frac{1}{2}$ as $r$ (bid to budget ratio) goes to zero. In adwords with concave returns studied in [8], $A_t$ is diagonal for all $t$ and $\psi$ is separable [2].

For any $p \ge 1$ let $\mathcal{B}_p$ be the $l_p$-norm ball. We can rewrite the penalty function $-\sum_{i=1}^n (u_i - 1)_+$ in the adwords objective using the distance from $\mathcal{B}_\infty$: we have $\sum_{i=1}^n (u_i - 1)_+ = d_1(u, \mathcal{B}_\infty)$, where $d_1(\cdot, C)$ is the $l_1$ norm distance from set $C$. For $p \in [1, \infty)$ the function $-d_1(u, \mathcal{B}_p)$ although not separable it satisfies Assumption 1. The proof is given in the supplementary materials.

**Examples on the positive semidefinite cone.** Let $K = S_+^n$ and note that $K^* = K$. Two examples that satisfy Assumption 1 are $\psi(U) = \log\det(U + A_0)$, and $\psi(U) = \mathbf{tr}U^p$ with $p \in (0, 1)$. We refer the reader to [10] for examples of online problems that entails $\log\det$ in the objective function and competitive ratio analysis of the simultanuous algorithm for these problems.

# 3  Smoothing of $\psi$ for improved competitive ratio

The technique of "smoothing" an (potentially non-smooth) objective function, or equivalently adding a strongly convex regularization term to its conjugate function, has been used in several areas. In convex optimization, a general version of this is due to Nesterov [17], and has led to faster convergence rates of first order methods for non-smooth problems. In this section, we study how replacing $\psi$ with a appropriately smoothed function $\psi_S$ helps improve the performance of the two algorithms discussed in section 1.1, and show that it provides optimal competitive ratio for two of the problems mentioned in section 2, adwords and online LP. We then show how to maximize the competitive ratio of both algorithms for a separable $\psi$ and compute the optimal smoothing by solving a convex optimization problem. This allows us to *design* the most effective smoothing customized for a given $\psi$: we maximize the bound on the competitive ratio over the set of smooth functions.(see subsection 3.2 for details).

Let $\psi_S$ denote an upper semi-continuous concave function (a smoothed version of $\psi$), and suppose $\psi_S$ satisfies Assumption 1. The algorithms we consider in this section are the same as Algorithms 1 and 2, but with $\psi$ replacing $\psi_S$. Note that the competitive ratio is computed with respect to the original problem, that is the offline primal and dual optimal values are still the same $P^\star$ and $D^\star$ as before.

From Lemma 1, we have that $D_{\text{sim}} \le \psi_S\left(\sum_{t=1}^m A_t\tilde{x}_t\right) - \psi^*(\tilde{y}_{m+1})$ and $D_{\text{seq}} \le \psi_S\left(\sum_{t=1}^m A_t\hat{x}_t\right) - \psi^*(\hat{y}_{m+1}) - \sum_{t=1}^m \langle A_t\hat{x}_t, \hat{y}_{t+1} - \hat{y}_t \rangle$. To simplify the notation, assume $\psi_S(0) = 0$ as before. Define

$$\alpha_{\psi,\psi_S} = \sup\{c \,|\, \psi^*(y) \ge \psi_S(u) + (c - 1)\psi(u), y \in \partial\psi_S(u), \ u \in K\}.$$

Then the conclusion of Theorem 1 for Algorithms 1 and 2 applied to the smoothed function holds with $\alpha_\psi$ replaced by $\alpha_{\psi,\psi_S}$.

## 3.1  Nesterov Smoothing

We first consider Nesterov smoothing [17], and apply it to examples on non-negative orthant. Given a proper upper semi-continuous concave function $\phi : \mathbb{R}^n \mapsto \mathbb{R} \cup \{-\infty\}$, let

$$\psi_S = (\psi^* + \phi^*)^*.$$

Note that $\psi_S$ is the supremal convolution of $\psi$ and $\phi$. If $\psi$ and $\phi$ are separable, then $\psi_S$ satisfies Assumption 1 for $K = \mathbb{R}_+^n$. Here we provide example of Nesterov smoothing for functions on non-negative orthant.

**Adwords:** The optimal competitive ratio for the Adwords problem is $1 - e^{-1}$. This is achieved by smoothing $\psi$ with $\phi^*(y) = \sum_{i=1}^m (y_i - \frac{e}{e-1})\log(e - (e - 1)y_i) - 2y_i$, which gives

$$\psi_{S,i}(u_i) - \psi_{S,i}(0) = \begin{cases} \frac{eu_i - \exp(u_i) + 1}{e - 1} & u_i \in [0, 1] \\ \frac{1}{e - 1} & u_i > 1, \end{cases}$$

## 3.2 Computing optimal smoothing for separable functions on $\mathbb{R}^n_+$

We now tackle the problem of finding the optimal smoothing for separable functions on the positive orthant, which as we show in an example at the end of this section is not necessarily given by Nesterov smoothing. Given a separable monotone $\psi(u) = \sum_{i=1}^n \psi_i(u_i)$ and $\psi_S(u) = \sum_{i=1}^n \psi_{S,i}(u_i)$ on $\mathbb{R}^n_+$ we have that $\alpha_{\psi,\psi_S} \geq \min_i \alpha_{\psi_i,\psi_{S,i}}$.

To simplify the notation, drop the index $i$ and consider $\psi : \mathbb{R}_+ \mapsto \mathbb{R}$. We formulate the problem of finding $\psi_S$ to maximize $\alpha_{\psi,\psi_S}$ as an optimization problem. In section 4 we discuss the relation between this optimization method and the optimal algorithm presented in [8]. We set $\psi_S(u) = \int_0^u y(s)ds$ with $y$ a continuous function ($y \in C[0,\infty)$), and state the infinite dimensional convex optimization problem with $y$ as a variable,

$$\begin{aligned} \text{minimize} \quad & \beta \\ \text{subject to} \quad & \int_0^u y(s)ds - \psi^*(y(u)) \leq \beta\psi(u), \qquad \forall u \in [0,\infty) \\ & y \in C[0,\infty), \end{aligned} \qquad (8)$$

where $\beta = 1 - \alpha_{\psi,\psi_S}$ (theorem 1 describes the dependence of the competitive ratios on this parameter). Note that we have not imposed any condition on $y$ to be non-increasing (i.e., the corresponding $\psi_S$ to be concave). The next lemma establishes that every feasible solution to the problem (8) can be turned into a non-increasing solution.

**Lemma 2** *Let $(y, \beta)$ be a feasible solution for problem (8) and define $\bar{y}(t) = \inf_{s \leq t} y(s)$. Then $(\bar{y}, \beta)$ is also a feasible solution for problem (8).*

In particular if $(y, \beta)$ is an optimal solution, then so is $(\bar{y}, \beta)$. The proof is given in the supplement. Revisiting the adwords problem, we observe that the optimal solution is given by $y(u) = \left( \frac{e - \exp(u)}{e - 1} \right)_+$, which is the derivative of the smooth function we derived using Nesterov smoothing in section 3.1. The optimality of this $y$ can be established by providing a dual certificate, a measure $\nu$ corresponding to the inequality constraint, that together with $y$ satisfies the optimality condition. If we set $d\nu = \exp(1-u)/(e-1) \, du$, the optimality conditions are satisfied with $\beta = (1 - 1/e)^{-1}$. Also note that if $\psi$ plateaus (e.g., as in the adwords objective), then one can replace problem (8) with a problem over a finite horizon.

**Theorem 2** *Suppose $\psi(t) = c$ on $[u', \infty)$ ($\psi$ plateaus). Then problem (8) is equivalent to*

$$\begin{aligned} \text{minimize} \quad & \beta \\ \text{subject to} \quad & \int_0^u y(s)ds - \psi^*(y(u)) \leq \beta\psi(u), \qquad \forall u \in [0,u'] \\ & y(u') = 0, \quad y \in C[0,u']. \end{aligned} \qquad (9)$$

So for a function $\psi$ with a plateau, one can discretize problem (9) to get a finite dimensional problem,

$$\begin{aligned} \text{minimize} \quad & \beta \\ \text{subject to} \quad & h \sum_{s=1}^t y[s] - \psi^*(y[t]) \leq \beta\psi(ht) \qquad \forall t \in [d] \\ & y[d] = 0, \end{aligned} \qquad (10)$$

where $h = u'/d$ is the discretization step. Figure 1a shows the optimal smoothing for the piecewise linear function $\psi(u) = \min(.75, \ u, \ .5u + .25)$ by solving problem (10). We point out that the optimal smoothing for this function is *not* given by Nesterov's smoothing (even though the optimal smoothing can be derived by Nesterov's smoothing for a piecewise linear function with only two pieces, like the adwords cost function). Figure 1d shows the difference between the conjugate of the optimal smoothing function and $\psi^*$ for the piecewise linear function, which we can see is not concave. We simulated the performance of the simultaneous algorithm on a dataset with $n = m$, $F_t$ simplex, and $A_t$ diagonal. We varied $m$ in the range from 1 to 30 and for each $m$ calculated the the smallest competitive ratio achieved by the algorithm over $(10m)^2$ random permutation of $A_1, \ldots, A_m$. Figure 1i depicts this quantity vs. $m$ for the optimal smoothing and the Nesterov smoothing. For the Nesterov smoothing we used the function $\phi^*(y) = (y - \frac{\sqrt{e}}{\sqrt{e}-1}) \log(\sqrt{e} - (\sqrt{e} - 1)y) - \frac{3}{2}y$.

In cases where a bound $u_{\max}$ on $\sum_{t=1}^m A_t F_t$ is known, we can restrict $t$ to $[0, u_{\max}]$ and discretize problem (8) over this interval. However, the conclusion of Lemma 2 does not hold for a finite horizon

and we need to impose additional linear constraints $y[t] \leq y[t-1]$ to ensure the monotonicity of $y$. We find the optimal smoothing for two examples of this kind: $\psi(u) = \log(1+u)$ over $[0, 100]$ (Figure 1b), and $\psi(u) = \sqrt{u}$ over $[0, 100]$ (Figure 1c). Figure 1e shows the competitive ratio achieved with the optimal smoothing of $\psi(u) = \log(1+u)$ over $[0, u_{\max}]$ as a function of $u_{\max}$. Figure 1f depicts this quantity for $\psi(u) = \sqrt{u}$.

### 3.3 Competitive ratio bound for the sequential algorithm

In this section we provide a lower bound on the competitive ratio of the sequential algorithm (Algorithm 1). Then we modify Problem (8) to find a smoothing function that optimizes this competitive ratio bound for the sequential algorithm.

**Theorem 3** *Suppose $\psi_S$ is differentiable on an open set containing $K$ and satisfies Assumption 1. In addition, suppose there exists $c \in K$ such that $A_t F_t \leq_K c$ for all $t$, then*

$$P_{\text{seq}} \geq \frac{1}{1 - \alpha_{\psi,\psi_S} + \kappa_{c,\psi,\psi_S}} D^\star,$$

*where $\kappa$ is given by*

$$\kappa_{c,\psi,\psi_S} = \inf\{r \mid \langle c, \nabla\psi_S(0) - \nabla\psi_S(u)\rangle \leq r\psi(u), u \in K\}$$

**Proof:** Since $\psi_S$ satisfies Assumption 1, we have $\hat{y}_{t+1} \leq_{K^*} \hat{y}_t$. Therefore, we can write:

$$\sum_{t=1}^{m} \langle A_t \hat{x}_t, \hat{y}_t - \hat{y}_{t+1} \rangle \leq \sum_{t=1}^{m} \langle c, \hat{y}_t - \hat{y}_{t+1} \rangle = \langle c, \hat{y}_0 - \hat{y}_{m+1} \rangle \qquad (11)$$

Now by combining the duality gap given by Lemma 1 with 11, we get $D_{\text{seq}} \leq \psi_S \left(\sum_{t=1}^{m} A_t \hat{x}_t\right) - \psi^*(\hat{y}_{m+1}) + \langle c, \nabla\psi_S(0) - \nabla\psi_S \left(\sum_{t=1}^{m} A_t \hat{x}_t\right)\rangle$. The conclusion follows from the definition of $\alpha_{\psi,\psi_S}$, $\kappa_{c,\psi,\psi_S}$ and the fact that $D_{\text{seq}} \geq D^\star$. $\square$

Based on the result of the previous theorem we can modify the optimization problem set up in Section 3.2 for separable functions on $\mathbf{R}_+^n$ to maximize the lower bound on the competitive ratio of the sequential algorithm. Note that when $\psi$ and $\psi_S$ are separable, we have $\kappa_{c,\psi,\psi_S} \leq \max_i \kappa_{c_i,\psi_i,\psi_{S_i}}$. Therefore, similar to the previous section to simplify the notation we drop the index $i$ and assume $\psi$ is a function of a scalar variable. The optimization problem for finding $\psi_S$ that minimizes $\kappa_{c,\psi,\psi_S} - \alpha_{\psi,\psi_S}$ is as follows:

$$\begin{aligned}
\text{minimize} \quad & \beta \\
\text{subject to} \quad & \int_0^u y(s)ds + c(\psi'(0) - y(u)) - \psi^*(y(u)) \leq \beta\psi(u), \qquad \forall u \in [0, \infty) \qquad (12) \\
& y \in C[0, \infty).
\end{aligned}$$

For adwords, the optimal solution is given by $\beta = \frac{1}{1 - \exp(-\frac{1}{c+1})}$ and $y(u) = \beta\left(1 - \exp\left(\frac{u-1}{1+c}\right)\right)_+$, which gives a competitive ratio of $1 - \exp\left(\frac{-1}{c+1}\right)$. In Figure 1h we have plotted the competitive ratio achieved by solving problem 12 for $\psi(u) = \log\det(1+u)$ with $u_{\max} = 100$ as a function of $c$. Figure 1g shows the competitive ratio as a function of $c$ for the piecewise linear function $\psi(u) = \min(.75, u, .5u + .25)$.

## 4 Discussion and Related Work

We discuss results and papers from two communities, computer science theory and machine learning, related to this work.

**Online optimization.** In [8], the authors proposed an optimal algorithm for adwords with differentiable concave returns (see examples in section 2). Here, "optimal" means that they construct an instance of the problem for which competitive ratio bound cannot be improved, hence showing the bound is tight. The algorithm is stated and analyzed for a twice differentiable, separable $\psi(u)$. The assignment rule for primal variables in their proposed algorithm is explained as a continuous process. A closer look reveals that this algorithm falls in the framework of algorithm 2, with the only difference being that at each step, $(\tilde{x}_t, \tilde{y}_t)$ are chosen such that

$$\begin{aligned}
& \tilde{x}_t \in \text{argmax}\langle x, A_t^T \tilde{y}_t \rangle \\
& \forall i \in [n]: \quad \tilde{y}_{t,i} = \nabla\psi_i(v_i(u_i)), \quad u_i = (\sum_{t=1}^{t} A_s \tilde{x}_s)_i,
\end{aligned}$$

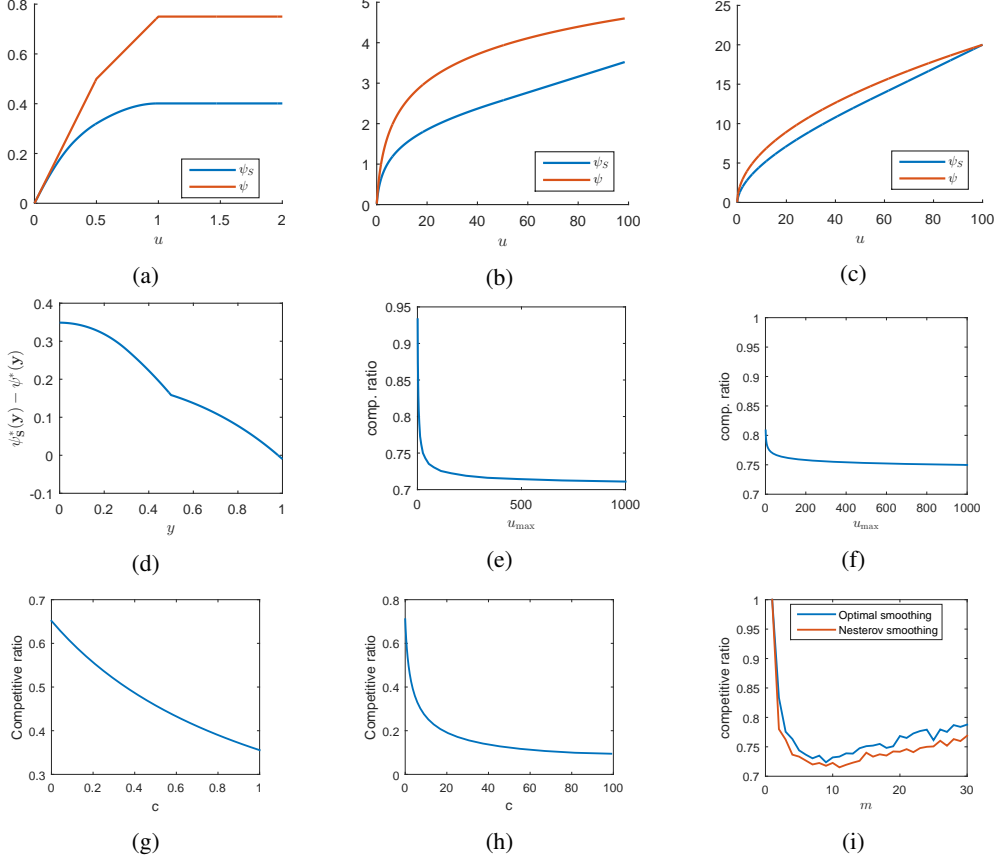

Figure 1: Optimal smoothing for $\psi(u) = \min(.75,\ u,\ .5u+.25)$ **(a)**, $\psi(u) = \log(1+u)$ over $[0, 100]$ **(b)**, and $\psi(u) = \sqrt{u}$ over $[0, 100]$ **(c)**. The competitive ratio achieved by the optimal smoothing as a function of $u_{\max}$ for $\psi(u) = \log(1+u)$ **(e)** and $\psi(u) = \sqrt{u}$ **(f)**. $\psi_S^* - \psi^*$ for the piecewise linear function **(d)**. The competitive ratio achieved by the optimal smoothing for the sequential algorithm as a function of $c$ for $\psi(u) = \min(.75,\ u,\ .5u + .25)$ **(g)** and $\psi(u) = \log(1 + u)$ with $u_{\max} = 100$ **(h)**. **i**, Competitive ratio of the simultaneous algorithm for $\psi(u) = \min(.75,\ u,\ .5u + .25)$ as a function of $m$ with optimal smoothing and Nesterov smoothing (see text).

where $v_i : \mathbb{R}_+ \mapsto \mathbb{R}_+$ is an increasing differentiable function given as a solution of a nonlinear differential equation that involves $\psi_i$ and may not necessarily have a closed form. The competitive ratio is also given based on the differential equation. They prove that this gives the optimal competitive ratio for the instances where $\psi_1 = \psi_2 = \ldots = \psi_m$.

Note that this is equivalent of setting $\psi_{S,i}(u_i) = \psi(v_i(u_i))$. Since $v_i$ is nondecreasing $\psi_{S,i}$ is a concave function. On the other hand, given a concave function $\psi_{S,i}(\mathbb{R}_+) \subset \psi_i(\mathbb{R}_+)$, we can set $v_i : \mathbb{R}_+ \mapsto \mathbb{R}_+$ as $v_i(u) = \inf\{z \mid \psi_i(z) \geq \psi_{S,i}(u)\}$. Our formulation in section 3.2 provides a *constructive* way of finding the optimal smoothing. It also applies to non-smooth $\psi$.

**Online learning.** As mentioned before, the dual update in Algorithm 1 is the same as in Follow-the-Regularized-Leader (FTRL) algorithm with $-\psi^*$ as the regularization. This primal dual perspective has been used in [20] for design and analysis of online learning algorithms. In the online learning literature, the goal is to derive a bound on *regret* that optimally depends on the horizon, $m$. The goal in the current paper is to provide competitive ratio for the algorithm that depends on the function $\psi$. Regret provides a bound on the duality gap, and in order to get a competitive ratio the regularization function should be crafted based on $\psi$. A general choice of regularization which yields an optimal regret bound in terms of $m$ is *not* enough for a competitive ratio argument, therefore existing results in online learning do not address our aim.

## Footnotes

[1]Also if the original problem is a convex relaxation of an integer program, meaning that each $F_t = \operatorname{conv}\mathcal{F}_t$ where $\mathcal{F}_t \subset \mathbb{Z}^l$, then $\hat{x}_t$ can always be chosen to be integral while integrality may not hold for the solution of the second algorithm.

[2]Note that in this case one can remove the assumption that $\partial\psi_i \subset \mathbb{R}_+$ since if $\tilde{y}_{t,i} = 0$ for some $t$ and $i$, then $\tilde{x}_{s,i} = 0$ for all $s \ge t$.

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
