[Supplementary Material]

# Supplementary Material for Designing smoothing functions for improved worst-case competitive ratio in online optimization

**Reza Eghbali**
Department of Electrical Engineering
University of Washington
Seattle, WA 98195
eghbali@uw.edu

**Maryam Fazel**
Department of Electrical Engineering
University of Washington
Seattle, WA 98195
mfazel@uw.edu

## 1 Proofs

**Proof of Lemma 1:** Using the definition of $D_{\mathrm{sim}}$, we can write:

$$
\begin{aligned}
D_{\mathrm{sim}} &= \sum_{t=1}^{m} \sigma_t(A_t^T \tilde{y}_t) - \psi^*(\tilde{y}_{m+1}) \\
&= \sum_{t=1}^{m} \langle A_t \tilde{x}_t, \tilde{y}_t \rangle - \psi^*(\tilde{y}_{m+1}) \\
&\leq \sum_{t=1}^{m} (\psi(\sum_{s=1}^{t} A_s \tilde{x}_s) - \psi(\sum_{s=1}^{t-1} A_s \tilde{x}_s)) - \psi^*(\tilde{y}_{m+1}) \\
&= \psi(\sum_{s=1}^{m} A_s \tilde{x}_s) - \psi(0) - \psi^*(\tilde{y}_{m+1}),
\end{aligned}
$$

where in the inequality follows from concavity of $\psi$, and the last line results from the sum telescoping. Similarly, we can bound $D_{\mathrm{seq}}$:

$$
\begin{aligned}
D_{\mathrm{seq}} &= \sum_{t=1}^{m} \sigma_t(A_t^T \hat{y}_t) - \psi^*(\hat{y}_{m+1}) \\
&= \sum_{t=1}^{m} \langle A_t \hat{x}_t, \hat{y}_t \rangle - \psi^*(\hat{y}_{m+1}) \\
&= \sum_{t=1}^{m} \langle A_t \hat{x}_t, \hat{y}_t - \hat{y}_{t+1} \rangle + \sum_{t=1}^{m} \langle A_t \hat{x}_t, \hat{y}_{t+1} \rangle - \psi^*(\hat{y}_{m+1}) \\
&\leq \sum_{t=1}^{m} \langle A_t \hat{x}_t, \hat{y}_t - \hat{y}_{t+1} \rangle + \sum_{t=1}^{m} (\psi(\sum_{s=1}^{t} A_s \hat{x}_s) - \psi(\sum_{s=1}^{t-1} A_s \hat{x}_s)) - \psi^*(\hat{y}_{m+1}) \\
&= \sum_{t=1}^{m} \langle A_t \hat{x}_t, \hat{y}_t - \hat{y}_{t+1} \rangle + \psi(\sum_{s=1}^{m} A_s \hat{x}_s) - \psi(0) - \psi^*(\hat{y}_{m+1}).
\end{aligned}
\tag{1}
$$

Figure 1: An example of $y(u)$ (solid blue) and $\bar{y}(u)$ (dashed red).

When $\psi$ is differentiable with Lipschitz gradient, we can use the following inequality that is equivalent to Lipschitz continuity of the gradient:

$$\psi(u') \geq \psi(u) + \langle \nabla \psi(u), u' - u \rangle - \frac{1}{2\mu} \|u - u'\|^2 \quad u, u' \in K,$$

to get

$$
\begin{aligned}
D_{\text{seq}} &= \sum_{t=1}^{m} \sigma_t(A_t^T \hat{y}_t) - \psi^*(\hat{y}_{m+1}) \quad (2) \\
&= \sum_{t=1}^{m} \langle A_t \hat{x}_t, \hat{y}_t \rangle - \psi^*(\hat{y}_{m+1}) \\
&\leq \sum_{t=1}^{m} \frac{1}{2\mu} \|A_t \hat{x}_t\|^2 + \sum_{t=1}^{m} (\psi(\sum_{s=1}^{t} A_s \hat{x}_s) - \psi(\sum_{s=1}^{t-1} A_s \hat{x}_s)) - \psi^*(\hat{y}_{m+1}) \\
&= \sum_{t=1}^{m} \frac{1}{2\mu} \|A_t \hat{x}_t\|^2 + \psi(\sum_{s=1}^{m} A_s \hat{x}_s) - \psi(0) - \psi^*(\hat{y}_{m+1}).
\end{aligned}
$$

□

**Proof of Lemma 2:** Let $(y, \beta)$ be a feasible solution for problem (8). Note that $y \geq 0$ since **dom** $\psi^* \subset \mathbb{R}_+$ by the fact that $\psi$ is non-decreasing. Let $\bar{y}(u) = \inf_{s \leq u} y(s)$. Note that $\bar{y}$ is continuous. Define

$$\beta(u) = \frac{\int_{s=0}^{u} y(s)\, ds - \psi^*(y(u))}{\psi(u)}, \qquad \bar{\beta}(u) = \frac{\int_{s=0}^{u} \bar{y}(s)\, ds - \psi^*(\bar{y}(u))}{\psi(u)},$$

with the definition modified with the right limit at $u = 0$. For any $u$ such that $\bar{y}(u) = y(u)$, we have:

$$\beta(u) = \frac{\int_{s=0}^{u} y(s)\, ds - \psi^*(y(u))}{\psi(u)} \geq \frac{\int_{s=0}^{u} \bar{y}(s)\, ds - \psi^*(y(u))}{\psi(u)} = \bar{\beta}(u).$$

Now, we consider the set $\{u \mid \bar{y}(u) \neq y(u)\}$. By the definition of $\bar{y}$, we have $\bar{y}(0) = y(0)$. Since both functions are continuous, the set $\{u \mid \bar{y}(u) \neq y(u)\}$ is an open subset of $\mathbb{R}$ and hence can be written as a countable union of disjoint open intervals. Specifically, we can define the end points of the intervals as:

$$
\begin{aligned}
a_0 &= b_0 = 0, \\
a_i &= \inf\{u > b_{i-1} \mid y(u) > \bar{y}(u)\}, \quad \forall i \in \{1, 2, \ldots\} \\
b_i &= \inf\{u > a_i \mid y(u) = \bar{y}(u)\}, \quad \forall i \in \{1, 2, \ldots\}.
\end{aligned}
$$

then $\{u \mid \bar{y}(u) \neq y(u)\} = \bigcup_{i \in \{1,2,\ldots\}} (a_i,\ b_i)$.(See Figure 1)

For any $i \in \{1, 2, \ldots\}$, we show that $\beta(u) \geq \bar{\beta}(u)$ on $(a_i,\ b_i)$. If $a_i = \infty$, then $b_i = \infty$, so we assume that $a_i < \infty$. By the definition of $a_i$ and $b_i$, $\bar{y}(u)$ is constant on $(a_i,\ b_i)$. Also, we have $y(a_i) = \bar{y}(a_i)$. Similarly, we have $y(b_i) = \bar{y}(b_i)$ whenever $b_i < \infty$.

Since $\bar{y}(u) \leq y(u)$ for all $u$ and $y(a_i) = \bar{y}(a_i)$, we have

$$\beta(a_i) \geq \bar{\beta}(a_i). \tag{3}$$

If $b_i < \infty$, similarly by the fact that $y(b_i) = \bar{y}(a_i) = \bar{y}(b_i)$, we have

$$\beta(b_i) \geq \bar{\beta}(b_i). \tag{4}$$

Now we consider the case where $b_i = \infty$. In this case we have $\bar{y}(u) = \bar{y}(a_i)$ on $(a_i, \infty)$. We consider two cases based on the asymptotic behavior of $\psi$. If $\lim_{u \to \infty} \psi(u) = +\infty$ ($\psi$ is unbounded), then we have

$$\limsup_{u \to \infty} \beta(u) = \limsup_{u \to \infty} \frac{\int_{s=0}^{u} y(s)\, ds}{\psi(u)} \geq \limsup_{u \to \infty} \frac{\int_{s=0}^{u} \bar{y}(a_i)\, ds}{\psi(u)} = \lim_{u \to \infty} \bar{\beta}(u). \tag{5}$$

Here we used the fact that $-\psi^*(y(u))$ is bounded. This follows from the fact $\psi^*$ is monotone thus:

$$-\psi^*(y(u)) \leq -\psi^*(\bar{y}(a_i)),$$

and $-\psi^*(\bar{y}(a_i)) < \infty$ because if $-\psi^*(\bar{y}(a_i)) = \infty$, then $\beta(a_i) \geq \bar{\beta}(a_i) = \infty$ which contradicts the feasibility of $(y, \beta)$.

Now consider the case when $\lim_{u \to \infty} \psi(u) = M$ for some positive constant $M$. In this case, $-\psi^* \leq M$. We claim that $y(a_i) = 0$ and $\liminf_{t \to \infty} y(u) = 0$. Suppose $\liminf_{u \to \infty} y(u) > 0$, then $\limsup_{u \to \infty} \beta(u) = \infty$ since the numerator in the definition of $\beta$ tends to infinity while the denominator is bounded. But this contradicts feasibility of $(y, \beta)$. On the other hand, by the definition of $a_i$ and $b_i$ we should have $y(a_i) = \bar{y}(a_i) \leq \liminf_{u \to \infty} y(u)$. Combining this with the fact that $\bar{y}(a_i) \in \mathbf{dom}\, \psi^* \subset \mathbb{R}_+$, we conclude that $y(a_i) = 0$. Using that $y(a_i) = 0$ and $\liminf_{u \to \infty} y(u) = 0$, we get:

$$
\begin{aligned}
\limsup_{u \to \infty} \beta(u) &= \limsup_{u \to \infty} \frac{\int_{s=0}^{u} y(s)\, ds - \psi^*(y(u))}{\psi(u)} \\
&\geq \lim_{u \to \infty} \frac{\int_{s=0}^{u} y(s)\, ds - \psi^*(0)}{M} \\
&\geq \frac{\int_{s=0}^{a_i} \bar{y}(s)\, ds - \psi^*(0)}{M} = \lim_{u \to \infty} \bar{\beta}(u),
\end{aligned} \tag{6}
$$

where in the last inequality we used the fact that $\bar{y}(u) = 0$ for $u \geq a_i$.

Let $\psi'$ be the right derivative of $\psi$. Since $\psi$ is concave, $\psi'$ is non-increasing. Therefore, the interval $(a_i, b_i)$ can be written as $(a_i, u'] \cup [u', b_i)$ such that $\psi'(u) \geq \bar{y}(a_i)$ on $(a_i, u']$ and $\psi'(u) \leq \bar{y}(a_i)$ on $[u', b_i)$. Since $\psi'(u) \geq \bar{y}(a_i)$ on $(a_i, u']$ we have:

$$
\begin{aligned}
\int_{a_i}^{u} \bar{y}(s)\, ds &= \int_{a_i}^{u} \bar{y}(a_i)\, ds \\
&\leq \int_{a_i}^{u} \psi'(s)\, ds = \psi(u) - \psi(a_i),
\end{aligned}
$$

for all $u \in (a_i, u']$. This yields:

$$
\begin{aligned}
\bar{\beta}(a_i) &= \frac{\int_{s=0}^{a_i} \bar{y}(s)\, ds - \psi^*(\bar{y}(a_i))}{\psi(u)} \\
&\geq \frac{\int_{s=0}^{a_i} \bar{y}(s)\, ds + \int_{a_i}^{u} \bar{y}(s)\, ds - \psi^*(\bar{y}(a_i))}{\psi(a_i) + \psi(u) - \psi(a_i)} = \bar{\beta}(u).
\end{aligned}
$$

for all $u \in (a_i, u']$. Here we used the fact that if $c_1 \geq c_2 > 0$ and $d_2 \geq d_1 \geq 0$, then

$$\frac{c_1}{c_2} \geq \frac{c_1 + d_1}{c_2 + d_2}.$$

Similarly, we have $\bar{\beta}(b_i) \geq \bar{\beta}(u)$ for any $u \in [u', b_i)$. Combining this with (3),(4),(5), and (6), we get:

$$
\begin{aligned}
\sup_{a_i \leq u \leq b_i} \bar{\beta}(u) &= \max(\bar{\beta}(a_i), \bar{\beta}(b_i)) \\
&\leq \max(\beta(a_i), \beta(b_i)) \leq \sup_{a_i \leq u \leq b_i} \beta(u).
\end{aligned}
$$

We conclude that $\bar{\beta}(u) \leq \beta(u)$ for all $t \geq 0$ hence $(\bar{y}, \beta)$ is a feasible solution for the problem.

**Proof of Theorem 2:** Let $(y, \beta)$ be a feasible solution for problem (8). By Lemma 2, we can assume that $y$ is non-increasing. First, note that $y \geq 0$ since $\mathbf{dom}\ \psi^* = [0,\ \infty)$. Define $\bar{y}(u) = y(u)$ for $u \leq u'$ and $\bar{y}(u) = 0$ for $u > u'$. We show that $(\bar{y}, \beta)$ is also a feasible solution for (8) modulo the continuity condition. Define

$$\beta(u) = \frac{\int_{s=0}^{u} y(s)\ ds - \psi^*(y(u))}{\psi(u)}, \qquad \bar{\beta}(u) = \frac{\int_{s=0}^{u} \bar{y}(s)\ ds - \psi^*(\bar{y}(u))}{\psi(u)}.$$

By the definition of $\bar{y}$, for all $u$, we have:

$$\int_0^u y(s)\ ds \geq \int_0^u \bar{y}(s)\ ds, \tag{7}$$

and $\beta(u) = \bar{\beta}(u)$ for $u \in [0,\ u']$. Since $y(u)$ is non-increasing and $y(u) \geq 0$, $\lim_{u \to \infty} y(u)$ exists. We claim that $\lim_{u \to \infty} y(u) = 0$. To see this note that if $\lim_{u \to \infty} y(u) > 0$, then

$$\lim_{u \to \infty} \int_{s=0}^{u} y(s)\ ds = \infty,$$

which contradicts the fact that $\beta(u) \leq \beta$ for all $u$. For all $u \geq u'$, now we have:

$$\sup_{u \geq u'} \beta(u) \geq \lim_{u \to \infty} \beta(u) = \frac{\lim_{u \to \infty} \int_{s=0}^{u} y(s)\ ds - \psi^*(0)}{\psi(u')}$$

$$\geq \frac{\int_{s=0}^{u'} \bar{y}(s)\ ds - \psi^*(0)}{\psi(u')} = \bar{\beta}(u'),$$

where the first equality follows from the fact that $\lim_{u \to \infty} y(u) = 0$, and in the last inequality, we used (7). Since $\bar{y}(u) = 0$ for $u > u'$, $\bar{\beta}(u)$ is constant on $[u'\ \infty)$. Therefore, $\sup_{u \geq u'} \bar{\beta}(u) = \bar{\beta}(u')$. Combining this with the previous inequality we get:

$$\sup_{u \geq u'} \beta(u) \geq \sup_{u \geq u'} \bar{\beta}(u).$$

Therefore, we conclude that $\bar{\beta}(u) \leq \beta$ for all $u$. Thus $(\bar{y}, \beta)$ is also a feasible solution for (8) modulo the continuity condition. Note that $\bar{y}(u)$ may not be continuous at $u'$. However, we can find a sequence of continuous functions $z^{(j)}$ that converge pointwise to $y$ and $z^{(i)}(u) = 0$ for all $i$ and $u \geq u'$. To do so we consider a sequence of real number $\epsilon_i \to 0$. We define $z^{(i)}(u) = \bar{y}(u)$ for $u \in [0,\ u' - \epsilon_i) \cup [u',\ \infty)$. On $[u' - \epsilon_i,\ u']$ we define $z^{(i)}(u)$ to be a linear function that take values $y(u' - \epsilon)$ and $0$ on the endpoints. Define

$$\beta_{z^{(i)}} = \sup_{u > 0} \frac{\int_{s=0}^{u} z^{(i)}(s)\ ds - \psi^*(z^{(i)}(u))}{\psi(u)}.$$

By upper semi-continuity of $\psi^*$, $\beta_{z^{(i)}}$ converges to $\bar{\beta}$.

Let $\beta^*$ be the optimal solution for problem (8). By the definition, there exits a feasible sequence $(y^{(j)}, \beta^{(j)})$ such that $\beta^{(j)}$ converges to $\beta^*$. Let $\bar{y}^{(j)}(u) = y^{(j)}(u)$ for $t \leq u'$ and $\bar{y}^{(j)}(u) = 0$ for $t > u'$. Note that $\bar{y}^{(j)}(u)$ may not be continuous at $u'$. However, we can find a sequence of continuous functions $(z^{(ji)},\ \beta_{z^{(ji)}})$ as in above. Now $\beta_{z^{(jj)}}$ converges to $\beta^*$.

$\square$

## 2   Distance from $l_p$ norm ball

In this section we prove that the function:

$$G(u) = -d_1(u, \mathcal{B}_p)$$

satisfies Assumption 1.

For any $u \in \mathbb{R}^n_+$, there exists $\bar{u} \in \mathcal{B}_p$ such that $d_1(u, \mathcal{B}_p) = \|u - \bar{u}\|_1$. the subdifferential of distance function is[1]:

$$\partial d_1(u, \mathcal{B}_p) = \partial \|u - \bar{u}\|_1 \cap N_{\mathcal{B}_p}(\bar{u}),$$

where $N_{\mathcal{B}_p}(u) = \{\xi \mid \langle \xi, v - u \rangle \geq 0, \ \forall v \in \mathcal{B}_p\}$ is the normal cone of $\mathcal{B}_p$ at $u$. In fact $d_1(u, \mathcal{B}_p) = \|u - \bar{u}\|_1$ if and only if $\partial \|u - \bar{u}\|_1 \cap N_{\mathcal{B}_p}(\bar{u}) \neq \varnothing$. When $u \in \text{int}\mathcal{B}_p$, $\bar{u} = u$ and $\partial d_1(u, \mathcal{B}_p) = \{0\}$. In order to find $\partial d_1(u, \mathcal{B}_p)$ when $u \notin \text{int}\mathcal{B}_p$, we first find $\bar{u}$ in this case. For any $r \geq 0$, define $u \wedge r \in \mathbb{R}^n_+$ to be:

$$(u \wedge r)_i = \min(u_i, r) \quad \forall i.$$

Note that $\|u \wedge 0\|_p = 0$ and $\|u \wedge (\max_i u_i)\|_p = \|u\|_p \geq 1$. Since $\|u \wedge r\|_p$ is a continuous function of $r$, by the intermediate value theorem, there exists $r_u \in (0, \max_i u_i]$ such that $\|u \wedge r_u\|_p = 1$. Now $\bar{u} = u \wedge r$. To see this note that:

$$\partial \|u - \bar{u}\|_1 \cap N_{\mathcal{B}_p}(\bar{u}) = \left\{ \frac{1}{r_u^{p-1}} (u \wedge r_u)^{\circ(p-1)} \right\} \quad \text{for} \quad r_u < \max_i u_i; \tag{8}$$

$$\partial \|u - \bar{u}\|_1 \cap N_{\mathcal{B}_p}(\bar{u}) = \left\{ \frac{z}{r_u^{p-1}} (u \wedge r_u)^{\circ(p-1)} \mid 0 \leq z \leq 1 \right\} \quad \text{for} \quad r_u = \max_i u_i; \tag{9}$$

where $\circ(p-1)$ denotes element-wise exponentiation. Now if $u' \leq u$, then $r_{u'} \leq r_u$ since $\|u' \wedge r\|_p \geq \|u' \wedge r\|_p$ for all $r$. Thus by (8) and (9), there exists $y \in \partial d_1(u, \mathcal{B}_p)$ such that $y \geq \partial d_1(u', \mathcal{B}_p)$.

## Footnotes

[1]For convex function we use $\partial$ to denote subdifferential.