[Reviews · NeurIPS 2016]

Reviewer 1

Summary

In this paper, the authors study a family of methods for online optimization in the (for lack of a better general descriptive term--if the authors know one, that would be great) generalized concave adwords problem. I am not as familiar with the area as I ought to be, so my review may be somewhat that of a neophyte. The general problem is to understand the difference between the online solution of the problem max. \psi(\sum_t A_t x_t) s.t. x_t \in F_t and its offline solution, where we measure quality by competitive ratio. The authors give a fairly clean analysis of the competitive ratio, based on convex analysis and the FTRL family of algorithms, and provide an extension to show how smoothing the function \psi can yield improved behavior.

Qualitative Assessment

I am pro-acceptance for this paper, though I have (of course) criticisms. Below are a number of comments I have about the paper, not particularly organized, but hopefully cogent enough. The approach via convex analysis/online regret/duality is cleaner than the literature I have seen on this problem, which focuses on the "continuous update" idea of Devanur and Jain ([DJ] from here)--frankly, a somewhat obtuse and confusing idea--so that using duality more carefully is nice. The main assumption on the superdifferentials relative to the cone inclusion, Assumption 1, certainly is important for the analysis, but I had a hard time understanding how much more generality it allowed over the standard separable case in [DJ]. The PSD cone is a nice example that makes clear that the generality is helpful, though I wonder if other natural examples exist. That would be quite helpful, as the notation in the paper is somewhat heavy, and I found constructing examples on the fly for intuition more challenging than I usually do. Because the algorithm in the simultaneous case is the same (I believe?) as the "continuous update" of [DJ], I thought that when specialized to linear problems one should attain a competitive ratio of 1 - 1/e rather than 1/2. Am I missing something? Is this a weakness in the analysis? Is there actually a gap? The smoothing appears to achieve 1 - 1/e, so some similar algorithm achieves the correct competitive ratio... I found this a bit unclear. A somewhat problematic issue with the paper is that the interaction of the smoothing operation with Assumption 1---the cone ordering condition on the superdifferential of \psi---is quite unclear. It appears that separable functions maintain good behavior on the positive orthant, which falls back into the [DJ] analysis. But the authors do not mention non-separable functions in Section 3, which I understood to be one of the major points of the current paper. Why the lack of discussion? To be clear, also, the "optimal" smoothing is chosen by optimization of a *bound* on the competitive ratio, not the ratio itself. A second issue with the paper is the lack of compelling examples. The plots of smoothed functions are fine, but I would really like to see an example of behavior of these operations on a real problem. Just because one has improved a bound does not always mean one has improved the algorithm. Frankly, I wish there were a "resubmission" period for these reviews, because this paper could use (1) a bit more space for clarity and (2) a few better experiments so that we could see the real effects of smoothing. I think there are nice ideas, but with a bit of touching up, they could become very nice. Minor comments: The authors never defined competitive ratio; it might be useful to have early on. I found the notation a tiny bit confusing, as I had to swap all standard notation from convex analysis to the concave case (e.g. convex vs. concave conjugate, etc.). I am not sure if this is avoidable, though the NIPS audience is more generally familiar with the convex case and it might be clearer to form the primal problem as minimization. Not a deal-breaker or anything. Pg. 4, line 112: calling D_seq the dual value was a little strange to me, as the sequence $\hat{y}_t$ does not obviously yield a dual feasible point, though as Theorem 1 shows D_sim and D_seq are actually tighter than D^*, the optimal dual value. It might be worth noting this earlier. Pg. 8, lines 284-end. A relevant reference for understanding regret vs. competitive ratio, at least in a restricted set of problems, might be Andrew et al.'s "Tale of Two Metrics" (http://users.cms.caltech.edu/~katrina/papers/twometrics.pdf).

Confidence in this Review

2-Confident (read it all; understood it all reasonably well)


Reviewer 2

Summary

The paper highlights the role of smoothing in designing algorithms for online resource allocation with low worst-case competitive ratio. The authors describe a framework that generalizes many of the known algorithms and provide an analysis based on smoothness properties of the objective. The main novel contribution is the possibility of computing the optimal smoothing for a given cost function. This is shown to match the optimal bounds for some problems, including adwords.

Qualitative Assessment

I was aware of the technical similarities between online algorithms minimizing competitive ratio and those minimizing regret, but I enjoyed reading the paper as it provides a very rigorous, yet self-contained, presentation of the connection between these methods, highlighting in particular the role of smoothing in designing low competitive-ratio algorithms. The idea of customising the smoothing function is very interesting and novel to the best of my knowledge. While I am excited about the technical content and novelty of the paper, I should state that this is above all a theoretical contribution, whose impact may be confined. I do not believe the paper showed novel competitive bounds for interesting problems. Moreover, the theoretical formalism of the paper makes it somewhat hard to digest and may not be a great fit for NIPS.

Confidence in this Review

2-Confident (read it all; understood it all reasonably well)


Reviewer 3

Summary

The paper analyses competitive ratios for two optimization algorithms, deriving bounds on the worst case ratio under specified conditions. This result is used to find smoothed cost functions.

Qualitative Assessment

This is a very specialized paper, of limited interest to the wider community. It contains no indication of its potential usefulness - although the abstract mentions areas such as matching, advertising, resource allocation, there is no discussion of how this research solves existing problems (or improves existing solutions). The analysis covers a subset of optimization problems (there are several assumptions made) without explaining the limitations involved in the results. I was also surprised to see no account of the complexity of this approach. The core of the paper is reasonably well-written but it lacks a proper introduction and conclusion (explaining why the problem should be solved, importance and usefulness of the solution, etc).

Confidence in this Review

2-Confident (read it all; understood it all reasonably well)


Reviewer 4

Summary

This paper introduces two primal-dual algorithms for a class of online convex optimization problems. Both try to greedily solve the analogous offline problem using the approximate information available at the current time, one sequentially and the other one simultaneously. By assuming that the gradient map is order-reversing, the authors are able to derive a lower bound on the duality gap of the problem and extend this to a bound on the competitive ratio. They then show how applying the same algorithms to smoothed versions of \psi can improve the competitive ratios, in particular deriving the optimal smoothing for separable functions on R_+^n by discretizing an infinite dimensional convex optimization problem.

Qualitative Assessment

This paper was well-organized and gave a complete picture on the problem that the authors were trying to address and how it fit within the overall landscape of the literature. They also built up to their main result systematically, which was appreciated. While the authors do discuss related work in both the beginning and end of the paper, it would be better to provide some direct comparisons between their algorithms and existing ones (where applicable) in terms of actual performance (either in the form of experiments or an in-depth discussion of computational complexity). I also have some technical questions that need to be clarified before I am willing to give a higher score: Main paper line 134-145: Why is the last inequality "...\geq D^*" true if the two points \tilde{y}_{m+1} and \tilde{y}_m on the LHS are different? Theorem 1, lines 127-136: How can P_{sim} depend on \tilde{x}, but \bar{\alpha}_{\psi} and D^* not? Can you carefully explain the application of Lemma 1 and this quantity to finish the proof on line 135? lines 197-201: I checked that these points are true, but it might be good to derive these arguments more carefully in the appendix. line 209-210: Where is the motivation for this choice of smoothing in the example? Appendix lines 35-36: Why is a_{i+1} > b_i? lines 43-44: Why are the first and last equalities in the display equation regarding \beta(t) and \bar{\beta}(t) true? lines 58-59: Should the last term be \beta(t) in the display equation?

Confidence in this Review

2-Confident (read it all; understood it all reasonably well)


Reviewer 5

Summary

The authors analyze the competitive ratio of two primal-dual algorithms on a class of conic online convex optimization problems that includes many important problems as special cases. They derive sufficient conditions on the objective function for the competitive ratio to be constant. Then, crucially, they show how these competitive ratios can be improved via smoothing. Unlike previous works, they show how to construct an optimal smoothing function for a problem instance via convex optimization.

Qualitative Assessment

The paper is extremely well-written, especially given the technical complexity of the work. The presentation is very precise and rigorous. The main idea of designing a smoothing function specific to each problem instance seems novel and important, and surprising to me, personally. I know this is a paper geared towards a more theoretical community, and so this is not necessarily expected, but I would have found it interesting to have a more detailed discussion of the practical advantages of the method on real problems, rather than the toy examples in the paper. I would have liked to get an idea of how much custom smoothing helps. For example, does it yield substantially better competitive ratios relative to e.g. Nesterov smoothing? However, I must add that I am not sufficiently familiar with the literature in this area to fully place the work into context.

Confidence in this Review

1-Less confident (might not have understood significant parts)